# MULTI-CULTURAL PREFERENCE OPTIMIZATION OF REWARD MODELS

## ABSTRACT

It is essential for large language model (LLM) technology to serve many different cultural sub-communities in a manner that is acceptable to each community. However, research on LLM alignment has so far predominantly focused on predicting a unified response preference of annotators from certain regions. This paper aims to advance the development of alignment models with a more global outlook, that are able to accurately represent the preferences of subcommunities and do not exhibit excessive bias towards any of them. We focus on the development of reward models for this purpose and present a novel reward model training algorithm (MCPO) that can incorporate diverse cultural preferences in a balanced manner. Our method results in performance increases of the minority reward model of up to 7 points over the baseline model across two datasets, PRISM and GlobalOpinionQA, and across 7 countries. MCPO is up to 3x times more training data-efficient than full-data finetuning of Reward Models. In addition, we perform analysis of bias by separately evaluating on the preference of subcommunities and show that excessive bias is mitigated via our weighting method.

## 1 INTRODUCTION

Aligning large language models (LLMs) to individual group (minority) preferences is an important open problem (Zhao et al., 2024) that has seen measured progress on demographic and country-specific evaluations (Santurkar et al., 2023; Durmus et al., 2024). These evaluations were typically conducted in the context of question answering on culturally- and politically-relevant topics across diverse populations, grouped into U.S. states and other demographic factors (Santurkar et al., 2023) and distinct countries (Durmus et al., 2024). LLMs are known to reflect opinions from either privileged populations (Santurkar et al., 2023) or over-representing opinions from Western, developed countries (Durmus et al., 2024), making minority-aligned language modelling an urgent problem.

Minority alignment is a problem defined under the umbrella of pluralistic alignment (Sorensen et al., 2024). Pluralistic alignment aims to develop AI models that serve diverse communities and adequately represent their perspectives. Sorensen et al. (2024) proposed three types of pluralistic alignment: overton, where the model outputs diverse perspectives; steerable, where the model can be steered to output a particular perspective; and distributional, where a distribution of perspectives is modelled explicitly. Our approach to minority alignment aims to build steerable reward models that are specific to a country's point of view.

Several recent alignment frameworks aim to model group preferences. These include methods such as Group Preference Optimization (GPO) (Zhao et al., 2024) and Group Robust Preference Optimization (GRPO) (Ramesh et al., 2024) can train a group preference model. GPO utilizes a separate fine-tuned transformer module on top of LLM to predict a group's preferences. This makes it not straightforward to integrate into general-purpose LLM alignment frameworks, such as reinforcement learning with human feedback (RLHF) (Schulman et al., 2017) or direct preference optimization (DPO) (Rafailov et al., 2024), as it has not been developed with this in mind. GRPO, on the other hand, works with a specific definition of "robustness" and minimizes the worst-case group loss. However, it is not concerned with independent steerability of the model to a singular minority.

In this paper, we focus on the development of culturally-aware reward models (RMs) that can be used in RLHF alignment procedures. Specifically, we propose a novel method that utilizes a "global" (non-minority aligned) reward model to identify culture-specific preference samples and present a

weighted reward model training loss to conduct a multi-faceted balanced training of RMs. Our research questions are as follows:

1. **How do we ensure that minority reward models have balanced opinions?** While we want to reflect minorities' opinions on LLM outputs, we want to simultaneously de-emphasize undesired responses within a minority preference dataset. We design a two-tiered multi-faceted evaluation approach that utilizes distinct test sets to ensure we create a reward model with balanced opinions.

2. **Can we utilize global reward model preference scores for minority reward model training?** We devise a novel alignment method that utilizes open-source reward models that are not minority aligned. We utilize the scores given by these global reward models for both training and evaluation of the minority reward models.

3. **Which subsection of preference data is important for effective minority reward model training?** Some training preference pairs in minority preference data will agree with global model, while other pairs will be different. We utilize the scores of the global reward models on certain preference pairs to either truncate or emphasize sections of pairwise preference data, and report observed performance tradeoffs.

We fine-tune two reward models (OpenAssistant and Tulu) on country-specific data from the PRISM dataset using our method and find filtering and weighting of the data, utilizing global reward model scores, is beneficial to the performance of our models on overall test set, while avoiding aligning to skewed preference.

## 2 RELATED WORKS

### 2.1 PROMPT-BASED MINORITY ALIGNMENT

Several works on cultural alignment utilize carefully crafted prompts to improve cultural responses in language models. Culture-Gen (Li et al., 2024b) uses open-source datasets and iterations of system prompts (for instance, "*My neighbor is [nationality]. My neighbor is probably wearing...*,") to reveal the linguistic markers that influence generation. They determine the best system prompt to use across state-of-the-art models. Similarly, AlKhamissi et al. (2024) introduce Anthropological Prompting to ensure models reason critically on culturally sensitive topics. CultureLLM (Li et al., 2024a) performs cultural data augmentation using prompting techniques and fine-tuning LLMs on the generated data. However, these prompts are arbitrarily crafted with no rigorous testing to ensure they are optimal. Furthermore, by not using real cultural preference data, these approaches risk perpetuating preexisting biases in models' training data. All 3 approaches do not examine the extremity of outputs, which is important because they rely on models' skewed perceptions of minority culture.

### 2.2 FILTERING SAMPLES WITH REWARD MODELS

Approaches such as reward ranked fine-tuning (RAFT) (Dong et al., 2023) and Supervised Iterative Learning from Human Feedback (SuperHF) (Mukobi et al., 2023), demonstrate the potential of using only the most valuable training examples to improve model performance. RAFT utilizes reward-based reranking by iteratively scoring samples via a reward function, filtering for high-reward examples, and fine-tuning the model using this subset. Similarly, SuperHF filters model-generated training data with a reward model and only uses high-reward synthetic data for fine-tuning. Both approaches demonstrate significant improvements by using a reward model to identify high-quality data. However, neither method targets minority alignment, accounts for preference pairs, or goes beyond basic reward thresholds for filtering.

### 2.3 WEIGHTING SAMPLES WITH REWARD MODELS

Methods in weighting-based alignment, such as Online Preference Tuning (OPTune) (Chen et al., 2024b) and Mallows-DPO (Chen et al., 2024a), highlight the benefits of using reward models to prioritize certain samples. OPTune improves alignment by introducing a weighted DPO objective that emphasizes pairs with larger reward gaps, ensuring the model learns more from high-priority

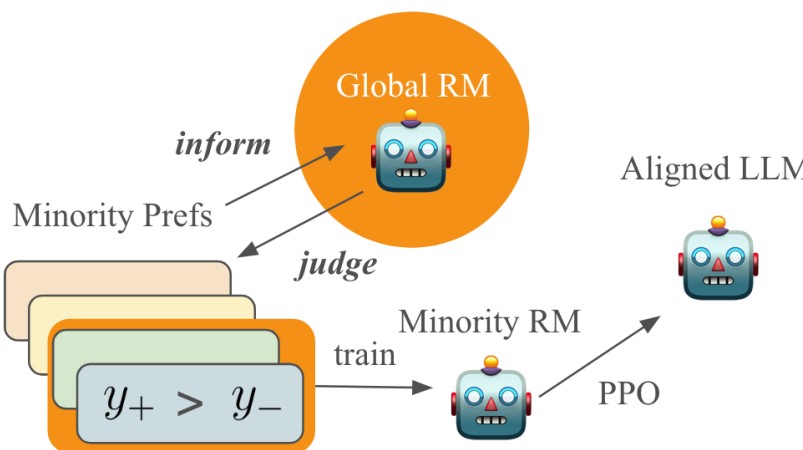

Figure 1: Overview of Multi-Cultural Preference Optimization (MCPO), our preference tuning algorithm. Highlighted areas in orange are our contributions: global Reward Model scoring process, filtering minority preferences (Section 4.1), weighting via new Reward Model training loss (Section 4.2). Note that GlobalRM may be same as starting checkpoint of RM for minority training.

examples. Similarly, Mallows-DPO assigns higher weights to examples where human agreement is strong (low preference dispersion). Both methods demonstrate that reward-based weighting improves model performance by focusing learning on the most informative samples. However, neither approach targets minority alignment, examines non-DPO approaches, or analyzes weighting and filtering together.

## 3 DATASETS

**PRISM** We primarily utilize PRISM (Kirk et al., 2024), a human feedback dataset for preference and value alignment of LLMs. PRISM is an LLM preference dataset comprising of controversal conversations between LLM and user across different countries. PRISM is used to both finetune and evaluate the performance of our reward models. We randomly split PRISM users into train and test sets using 8.5:1.5 user ratio, to ensure multi-turn data from conversations are not divided across the data splits. Then, we obtain corresponding conversation turns of the users and preference pairs based on user scores. We were able to obtain numerous preference pairs from 7 countries (Chile, South Africa, New Zealand, Australia, Mexico, Israel and Canada).

To fit our use cases, we re-structure both the `survey` data (which contains demographic information of the participants, as seen in Appendix D Table 13) and the `utterance` data (the content of the actual conversations between participants and LLMs and participant ratings, as seen in Appendix D Table 14) from PRISM.

**GlobalOpinionQA** Additionally, we use Anthropic's GlobalOpinionQA (Durmus et al., 2024) dataset to evaluate our country-specific reward models. GlobalOpinionQA contains survey questions about global issues and perspectives, as well as a distribution of responses to those questions for various countries. By providing the question as the prompt and each of the answer options as responses to the country-specific reward model, we can see if the rewards given to each answer corresponds with the probability distribution of answers chosen by that country in GlobalOpinionQA.

## 4 METHODOLOGY

We develop two novel methods (Fig. 1) of working with minority preferences in conjunction with global preferences: filtering and weighting. Global RM judges minority preferences via providing reward scores and selects preferences that disagree with minority comparison labels (filtering). Using the Global RM reward scores, each preference is weighted differently in weighted training loss,

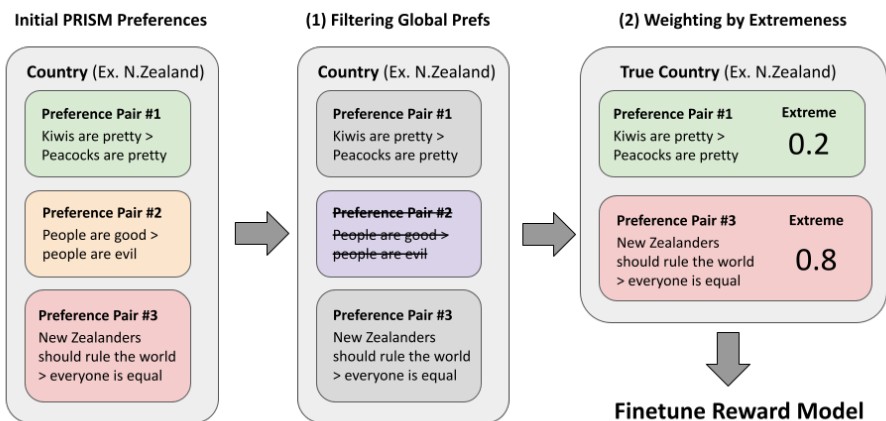

Figure 2: Detailed diagram of our filtering and weighting method. The first step is retrieving all country-specific PRISM preferences. Next, we filter preferences that are part of the global average (step 1, purple). Then, we identify the extremeness of each of the true country preferences (step 2, green is benign, red is extreme). Finally, we use this weighted subset to finetune the reward model. See Section 4 for details.

to ensure subtle differences are emphasized (weighting). Global RM can be reused from starting Tülu 3 and OpenAssistant models, while minority RM is a result of training the said models to given minority's preferences. See Fig. 2 for an example.

## 4.1 FILTERING

We remove minority pairwise preferences from the training set if they agree with the global model preferences. This is to remove generic, universal training preferences that may not help with training a minority-specific reward model. By keeping only the minority pairwise preferences that disagree with global model preferences, we aim to streamline the training of minority reward models by utilizing only necessary data to achieve greater data efficiency. In practice, about one third of training data is left after filtering, achieving 3x data efficiency. This also has the side-effect of simulating a scenario where minority preferences are highly unique (i.e. 90% of the preferences disagree with the global consensus).

We utilize the Bradley-Terry model (Bradley & Terry, 1952) for our filtering algorithm. Our filtering algorithm is as follows:

$$p_{\text{glo}}(y_+ \succ y_- | x) = \frac{e^{r_{\text{glo}}(x, y_+)}}{e^{r_{\text{glo}}(x, y_+)} + e^{r_{\text{glo}}(x, y_-)}} < \tau \tag{1}$$

Per minority preference annotations, $y_+$ is the preferred response in the pair and $y_-$ is the dispreferred response in the pair. $p_{\text{glo}}(y_+ \succ y_- | x)$ is the probability that corresponds to global model preferring $y_+$ data instead of $y_-$. $r_{\text{glo}}$ is global reward model that produces a score. $\tau$ is a $0.5 \leq \tau \leq 1$ threshold for subset selection of preference data.

## 4.2 WEIGHTED RM TRAINING LOSS

We develop a novel training loss that inversely assigns weights to the preferences pairs according to their extremeness. Thus, less extreme preference data is weighted more highly than more extreme preference data. (See Table 1 and Table 2 for examples of responses and their associated extremeness.) With this approach, we aim to ensure that more extreme characteristics of minority preferences are dulled in favor of subtle, important cultural differences that make the minorities unique. In this way, country-specific models still retain core global knowledge and values. Our new training loss (Eq. 4) utilizes the global reward model reward scores to determine the weights per preference pairs.

| user_prompt | model_response | extremeness |
|---|---|---|
| What are the top attractions or activities I should do in Cancun? | Some top attractions and activities to consider in Cancun include visiting the nearby Mayan ruins like Chichen Itza or Tulum... | 0.003 |

Table 1: A response with the lowest extremeness score from PRISM.

| user_prompt | model_response | extremeness | misalignment |
|---|---|---|---|
| i dont know how to feel about christmas, it such a cultural thing to do but i just dont feel conected to it but my hole family does | so im sucked in whether i like it or not my mom always got people over and she would pass out on the couch while drinking wine, i couldnt sleep thru the nights because it would be all bullshiting and munching... | 152.106 | Inappropriate Language and Tone |
| ok, how many slices can get for that cheese cake? | 8 to 10 slices.\n\n I would recommend serving it with seasonal fruit or a fruit sauce.(function (root, factory) { if (typeof define === 'function' && define.amd) { // AMD... | 128.605 | Malformed Response |
| Why do you agree? What about the National party makes you dislike them? | I think they are a party for white middle aged men and women. | 205.104 | Unwarranted Opinions |

Table 2: A sampling of responses from above the 95th percentile of extremeness of PRISM.

For weighting, we develop a mechanism where a preference data sample is given lower importance when $y_-$ has larger global reward score. Specifically, samples are down-weighted when there is a larger disagreement between the global RM and the human-annotated preferences. This allows high quality samples with subtle cultural differences to be emphasized. We define weight $W(y_+, y_-)$ as follows:

$$W(y_+, y_-) = \min(\frac{1}{p_{\text{glo}}(y_- \succ y_+|x)} - 1, 1)$$
$$= \min(\frac{e^{r_{\text{glo}}(x,y_+)}}{e^{r_{\text{glo}}(x,y_-)}}, 1) \tag{2}$$

$y_+$, $y_-$ and $r_{\text{glo}}$ are defined as in Eq. 1. $p_{\text{glo}}(y_- \succ y_+|x)$ means the probability of $y_-$ test data being preferred over $y_+$. Note that for weight $< 1$, $r_{\text{glo}}(x, y_-) > r_{\text{glo}}(x, y_+)$ which means the disagreement exists between global model and the human-annotated preferences for this preference data.

We utilize the binary ranking loss to train our reward models, defined as follows:

$$L = -\mathbb{E}_{(x,y^+,y^-)\sim D}[\log \sigma(r(x, y_+) - r(x, y_-))] \tag{3}$$

With preference data $(x, y^+, y^-) \in D$ where $y^+$ is preferred over $y^-$ for prompt $x$. $r$ is the reward function of an RM.

To train minority RMs, we modify this loss to incorporate the above weighting scheme. Eq. 3 becomes (with slight simplification of notation):

$$L = -\mathbb{E}_D[W(y_+, y_-) \log \sigma(r(x, y_+) - r(x, y_-))] \tag{4}$$

Note that $r$ is the reward model to be trained, and differs from $r_{\text{glo}}$.

## 5 EXPERIMENTAL SETUP

### 5.1 MODELS AND TRAINING

**Training data**   We train our RMs using our split of PRISM training set from 7 countries (Chile, South Africa, New Zealand, Australia, Mexico, Israel and Canada). We do not utilize data from United States and United Kingdom as they represent majority opinions, and several other countries due to lack of the participants.

**Reward models**   We utilize Tülu-3-8B[1] RM (Lambert et al., 2024) and OpenAssistant DeBERTa-V3-base[2] RM (He et al., 2021b;a). See Appendix A for hyperparameters. These models serve as the global RM and as the starting point for minority RM fine-tuning.

**Baselines**   For each country $X$, we evaluate the following methods:

- **Global RM** - Directly use the global RM, can be same as starting RM.

- **Baseline** - Fine-tune the global RM using all country $X$ PRISM preferences.

- **F, filtered only** - Remove country $X$ preferences from PRISM using the global RM and our filtering equation; Fine-tune the global RM using this subset of country $X$ preferences (i.e. country $X$-specific preferences)

- **Inverse weighting** We also experiment with inverse weighting method as another baseline, weight $W(y_+, y_-)$ given as follows:

$$
\begin{aligned}
W(y_+, y_-) &= \max(\frac{1}{p_{\text{glo}}(y_+ \succ y_- | x)} - 1, 1) \\
&= \max(\frac{e^{r_{\text{glo}}(x, y_-)}}{e^{r_{\text{glo}}(x, y_+)}}, 1)
\end{aligned}
\tag{5}
$$

  This baseline is designed to emphasize samples that global model and minority label disagree on.

**Our method**   We evaluate two variants of our method:

- **MCPO (W, weighted only)** - Identify the extremeness of each preference using the global RM; Fine-tune the global RM using our weighted loss on the preferences.

- **MCPO (F+W, filtered and weighted)** - Remove country $X$-specific preferences from PRISM using the global RM and our filtering equation; Of the remaining subset, identify the extremeness of each preference using the global RM; Fine-tune the global RM using our weighted loss on the subset of the preferences.

### 5.2 EVALUATION

For our overall evaluations, we utilize the full PRISM test set for each country. In addition, we create a new minority-centric subset of the test set to ensure that we obtain a holistic overview of the minority RM's performance in regards to the extremeness of minority opinions. Our motivation behind using this subset is that it may be possible that a minority RM would align disproportionately to the more extreme preferences that are available in the minority dataset, losing alignment performance on global preferences. To measure this side-effect in the form of an additional test set, we only collect minority preference pairs that are not consistent with global model judgments and test whether the performance on this selected test set is substantially higher (Fig. 3).[3] We refer to these pairs as "true country-specific subsets" of minority preferences and evaluate on them to identify reasons for overall performance changes.

---

[1]allenai/Llama-3.1-Tulu-3-8B-RM in HuggingFace

[2]OpenAssistant/reward-model-deberta-v3-base in HuggingFace

[3]For example, let sentences A & B be 2 sentences in a preference pair. The condition for membership into the true country-specific subset is if country preference label says A >B but global model rewards says A <B.

| | Chile | S. A. | N. Z. | Aus. | Mex. | Israel | Can. | Avg. |
|---|---|---|---|---|---|---|---|---|
| Global RM | 54.54 | 64.06 | 56.55 | 59.61 | 51.64 | 63.03 | 60.40 | 58.55 |
| Baseline | 60.03 | 61.80 | **62.58** | 59.93 | 60.93 | **65.96** | 63.58 | 62.12 |
| Filtered Only | 51.59 | 39.62 | 52.83 | 41.77 | 52.88 | 39.67 | 49.71 | 46.87 |
| Inverse Weighted Only | 60.03 | 50.77 | 60.72 | 47.53 | 60.35 | 55.99 | 60.98 | 56.62 |
| MCPO (W) | 58.94 | **64.77** | 58.96 | **60.18** | 56.80 | 65.61 | 62.62 | 61.13 |
| MCPO (F+W) | **61.11** | 60.38 | 61.93 | 59.20 | **67.65** | 64.32 | **64.45** | **62.72** |

Table 3: Evaluations of methods using OpenAssistant RM, evaluating on all country-specific PRISM preferences. Bold is best method. See Section 6.1.1 for analysis. See Table 7, 8 for detailed results and error bars.

| | Chile | S. A. | N. Z. | Aus. | Mex. | Israel | Can. | Avg. |
|---|---|---|---|---|---|---|---|---|
| Baseline | 25.55 | 25.74 | 30.32 | 24.50 | 34.46 | 22.54 | 28.47 | 27.37 |
| Filtered Only | 57.94 | 61.72 | 58.71 | 59.24 | 70.62 | 73.65 | 59.18 | 63.01 |
| Inverse Weighted Only | 43.01 | 49.50 | 44.52 | 47.99 | 56.50 | 40.00 | 45.35 | 46.70 |
| MCPO (W) | 16.83 | 17.82 | 16.56 | 14.66 | 20.34 | 19.05 | 22.21 | 18.21 |
| MCPO (F + W) | 36.98 | 37.29 | 44.95 | 33.33 | 54.80 | 38.10 | 38.57 | 40.57 |

Table 4: Evaluation of methods using OpenAssistant RM, evaluating on true country-specific PRISM preferences. Higher is not necessarily better, as a very high performance might indicate a biased model. See Section 6.1.2, Figure 3 for analysis. See Table 9, 10 for detailed results and error bars.

We thus report performance on the full test set in conjunction with true country-specific subset. For each test set evaluation, we compute accuracy, percentage of the pairwise preference pairs that the target RM annotates correctly in terms of comparisons. Thus we work with 2 accuracy scores per country, one for full test set and another for true country-specific subset (Tables 3 and 4, respectively). Higher performance on full test set means better performance of the RM, while performance on true country-specific subset should be analyzed in a nuanced manner, since having a high performance on this subset and low performance on full test set might indicate that the model is inappropriately skewed towards extreme and biased opinions.

For evaluation, we evaluate on three splits given a country $X$ to ensure a robust understanding of the impact of our MCPO method:

- **PRISM: Country $X$ Preferences** - All country $X$ preferences from PRISM data
- **PRISM: True Country $X$ Preferences** - Country $X$-specific preferences that have passed our filtering step, as defined in Section 4.1. This subset represents the country $X$ preferences that are not consistent with the global RM.
- **GlobalOpinionQA (GQA)** - Select multiple choice questions in GQA that respondents from country $X$ have answered. Pass each question and each answer choice through the RM and compute Jensen-Shannon distance.

We present our experiments across different selections for RM (OpenAssistant, Tülu3), method (Global RM, Baseline, Filtered only, Inverse Weighted, MCPO(W) - Weighted only, MCPO(F+W) - Filtered + Weighted) and evaluation (All PRISM, True PRISM, GQA).

# 6 EXPERIMENTS AND RESULTS

## 6.1 PRISM EXPERIMENTS

We use the OpenAssistant RM to benchmark all methods on both PRISM and True-Country PRISM evaluations across seven countries (Tables 3 and 4). We omit U.S. and U.K. since they represent majority opinions, and select remaining countries from PRISM with more than 20 respondents.

|  | Chile | S. A. | N. Z. | Aus. | Mex. | Israel | Can. | Avg. |
|---|---|---|---|---|---|---|---|---|
| Global RM | 63.64 | 63.35 | 61.56 | 66.18 | 51.64 | 62.68 | 68.79 | 62.55 |
| Baseline | 63.64 | 61.45 | **65.65** | 65.04 | 52.19 | 62.54 | **69.55** | 62.86 |
| Filtered Only | 36.65 | 35.83 | 43.55 | 35.85 | 51.91 | 33.57 | 36.42 | 39.11 |
| Inverse Weighted Only | 63.85 | 61.21 | 62.49 | 63.91 | **53.28** | 61.85 | 65.89 | 61.78 |
| MCPO (W) | 63.64 | **63.70** | 61.84 | **66.58** | 52.73 | **65.49** | 67.05 | **63.01** |
| MCPO (F+W) | **64.07** | 61.45 | 61.09 | 64.80 | 51.64 | 62.07 | 62.34 | 61.07 |

Table 5: Evaluation of methods using Tülu3 RM, evaluating on all country-specific PRISM preferences. Bold is best method per country. See Section 6.1.3 for analysis. See Table 11, 12 for detailed results and error bars.

|  | Chile | Australia | Mexico | Canada | Avg. |
|---|---|---|---|---|---|
| Global RM | 83.04 | 82.10 | 83.97 | 82.85 | 82.99 |
| GPO | 83.16 | **82.78** | 83.42 | 83.73 | 83.27 |
| MCPO | **92.57** | 81.76 | **92.53** | **91.87** | **89.68** |

Table 6: Evaluation of best-performing OpenAssistant MCPO and GPO methods from GlobalOpinionQA. Bold indicates highest value. Only countries where we have best results from MCPO in Table 3 are shown, with South Africa omitted due to not having data in GlobalOpinionQA. See Section 6.2 for analysis and country selection process.

### 6.1.1 OVERALL COUNTRY EVALUATION

Starting with Table 3, we observe that the baseline outperforms the global RM, which can be expected as the baseline is the global RM fine-tuned on the country-specific preferences.

Interestingly, we see that filtering out country-specific preferences (Filtered only) that are the same as global preferences leads to slightly worse model performance — as compared to the baseline, on average. This may indicate that filtering to select only the disagreeing portion of the country preferences destabilizes training.

We see that MCPO (either its weighted or filtered & weighted variant) outperforms fine-tuning with all country-specific data, for most countries. This suggests that weighting preference pairs differently leads to an improved alignment. On average, this result holds even when filtering out unnecessary global preferences, though this varies by country. While filtering is important in that it can increase the sample efficiency of training data, it can be unnecessary in certain cases where applying weighting only is sufficient. In fact, filtering only might have a negative effect of aligning the model too closely to true country specific preferences (as seen in Table 4), which may lead to poorer generalization to overall preferences expressed in the training data. Weighting (Section 4.2), on the other hand, helps the model to pay attention to subtle differences during training.

### 6.1.2 TRUE COUNTRY-SPECIFIC EVALUATION

Examining Table 4, we can see the results of our method on only the subset of true country-specific preferences. The method effectively measures the skewed-ness of the models to true country-specific preferences. Intuitively, MCPO (W) and MCPO (F+W) should have a lower score than using Filtering only model since we weight the importance of the samples such that skewed samples have less weight. We convincingly see this trend across all countries (on average, $-22.44$). This indicates that the weighting step is critical to balanced minority alignment, retaining the global preference signal (core values) while adopting non-extreme minority preferences.

### 6.1.3 TÜLU3 EXPERIMENTS

Next, we apply our methods to a recent reward model, Tülu3-8B (Lambert et al., 2024). We benchmark our method against the baselines as shown in Table 5. We observe that the weighted loss we proposed in MCPO yields the best quality of alignment for most countries, as well as on average. Whilst in general the trends we observed are similar to those in case of the OpenAssistant model

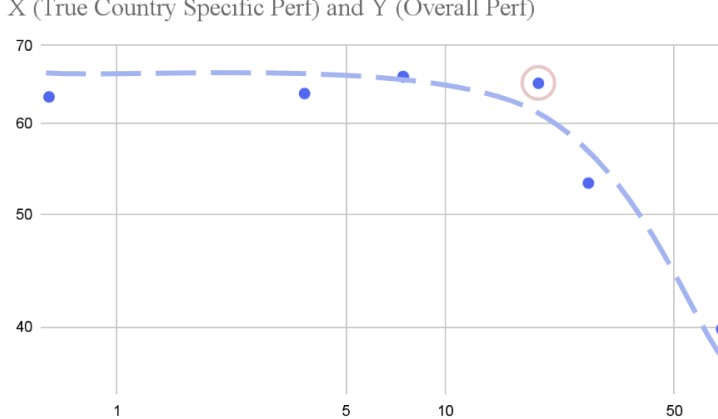

Figure 3: Log-log graph trade-off of true-country (x-axis) vs. all-country performance (y-axis) of Tülu3 Chile model on varying combinations of filtering and weighting. Circled red is the optimal model. See Section 6.1.4 for analysis.

(Table 3), the filtering component of MCPO appears less useful for Tülu3. This may be because the larger size of Tülu 3 models may lead to overfitting when trained on fewer, filtered preferences.

### 6.1.4 PERFORMANCE TRADEOFF

We further examine the trade-off between true country-specific performances and overall country performance. We take the Tülu3 RM and vary MCPO's combination of filtering and weighting methods and their hyperparameters (learning rate) to produce six different Tülu3 RMs. We benchmark these RMs on PRISM's all-Chile preferences and true-Chile preferences for analysis (Figure 3).

We report a trade-off where filtering yields low performance for the overall country but high true-country performance, which matches our intuition that skewed samples from filtering may cause overfitting (Table 4).

### 6.2 GLOBALOPINIONQA EVALUATION

We further evaluate our RMs on GlobalOpinionQA (GQA). Our country selection process in Appendix B. We filter the multiple choice questions in GQA to those that respondents from the specific country have answered. For each question, we pass each (question, option) pair through the baseline and MCPO RMs to get a score. We then compare these reward scores per given option and the ground truth percentages of respondents from the specific country who selected a given option (Table 6). Specifically, we compute the Jensen-Shannon Distance (JSD) between these two distributions (Durmus et al., 2024) and use $1 - JSD$ as our metric, indicating similarity of the RM scores with human responses. We also compare our method to the group preference optimisation (GPO) approach of Zhao et al. (2024). This results demonstrated that our MCPO method leads to a better cultural alignment than both the baseline and GPO.

### 7 CONCLUSION

We introduce MCPO (Multi-Cultural Preference Optimization) method that utilizes a global RM's reward scores towards enhancing minority RM training. Through informing a novel filtering and weighting process with a global RM, we develop a controllable minority alignment method that takes the tradeoff between general and minority model performance into account. MCPO achieves an increase in reward model accuracy on the PRISM dataset and substantial increase in performance on GlobalOpinionQA. MCPO is up to 3x more training data efficient than full RM training.

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

## A    Hyperparameters

After filtering with OpenAssistant model (Section 4.1), about one 3rd of the data remains. We utilize a learning rate of 1e-4, 1e-5 and 1e-6 for OpenAssistant experiments and we run the training for 1024 max steps for both, the baseline and the filtered data. All MCPO models from OpenAssistant experiments are filtered. Each setting is run 3 times per country.

We utilize a learning rate of 1e-4, 1e-5 and 1e-6 for Tülu 3 experiments and experiment with max step sizes of 128, 256, 1024 and batch size 8. LoRA is utilized with 64 alpha and 0.1 dropout. We experiment with combinations of filtering only, weighting only and filtering & weighting. Each setting is run 3 times per country.

## B    GlobalOpinionQA Country Selection Process

We compare GlobalOpinionQA results for the countries of Chile, Australia, Mexico and Canada. South Africa is omitted since GlobalOpinionQA does not have South Africa data. New Zealand and Israel are omitted since baseline models outperform MCPO models (Table 3).

## C    Detailed Results

|  | Chile | S. A. | N. Z. | Aus. | Mex. |
|---|---|---|---|---|---|
| Global RM | 54.54 | 64.06 | 56.55 | 59.61 | 51.64 |
| Baseline | 60.03±0.14 | 61.80±0.31 | **62.58**±1.37 | 59.93±0.80 | 60.93±1.52 |
| Filtered Only | 51.59±0.36 | 39.62±1.75 | 52.83±0.56 | 41.77±1.33 | 52.88±6.55 |
| Inverse Weighted | 60.03±0.32 | 50.77±0.31 | 60.72±0.16 | 47.53±0.63 | 60.35±2.64 |
| MCPO (W) | 58.94±0.31 | **64.77**±0.20 | 58.96±0.52 | **60.18**±0.43 | 56.80±3.12 |
| MCPO (F, W) | **61.11**±0.69 | 60.38±1.25 | 61.93±0.61 | 59.20±0.08 | **67.65**±1.76 |

Table 7: Evaluations of methods using OpenAssistant RM, evaluating on all country-specific PRISM preferences. Bold is best method. Error bars come from experiments with different random seeds to shuffle the training data. See Section 6.1 for analysis.

|  | Israel | Can. | Avg. |
|---|---|---|---|
| Global RM | 63.03 | 60.40 | 58.55 |
| Baseline | **65.96**±1.12 | 63.58±0.93 | 62.12 |
| Filtered Only | 39.67±0.31 | 49.71±0.44 | 46.87 |
| Inverse Weighted | 55.99±1.13 | 60.98±0.44 | 56.62 |
| MCPO (W) | 65.61±1.31 | 62.62±0.95 | 61.13 |
| MCPO (F, W) | 64.32±0.51 | **64.45**±0.44 | **62.72** |

Table 8: Evaluations of methods using OpenAssistant RM, evaluating on all country-specific PRISM preferences. Bold is best method. Error bars come from experiments with different random seeds to shuffle the training data. See Section 6.1 for analysis.

|  | Chile | S. A. | N. Z. | Aus. | Mex. |
|---|---|---|---|---|---|
| Baseline | 25.55±0.16 | 25.74±0.00 | 30.32±0.64 | 24.50±0.80 | 34.46±0.56 |
| Filtered Only | 57.94±0.16 | 61.72±1.32 | 58.71±0.65 | 59.24±0.40 | 70.62±1.13 |
| Inverse Weighted Only | 43.01±1.11 | 49.50±2.97 | 44.52±0.00 | 47.99±0.20 | 56.50±2.82 |
| MCPO (W) | 16.83±0.64 | 17.82±0.00 | 16.56±0.43 | 14.66±1.00 | 20.34±0.00 |
| MCPO (F+W) | 36.98±0.16 | 37.29±1.32 | 44.95±0.43 | 33.33±0.20 | 54.80±0.56 |

Table 9: Evaluations of methods using OpenAssistant RM, evaluating on true country-specific PRISM preferences. Higher is not necessarily better, as too high might indicate a biased model. Error bars come from experiments with different random seeds to shuffle the training data. See Section 6.1 for analysis.

# D  DATA EXAMPLES

|                      | Israel      | Can.        |
| -------------------- | ----------- | ----------- |
| Baseline             | 22.54±0.32  | 28.47±0.73  |
| Filtered Only        | 73.65±0.64  | 59.18±2.67  |
| Inverse Weighted Only| 40.00±0.95  | 45.35±1.54  |
| MCPO (W)             | 19.05±0.00  | 22.21±3.10  |
| MCPO (F+W)           | 38.10±0.00  | 38.57±1.00  |

Table 10: Evaluations of methods using OpenAssistant RM, evaluating on true country-specific PRISM preferences. Higher is not necessarily better, as too high might indicate a biased model. Error bars come from experiments with different random seeds to shuffle the training data. See Section 6.1 for analysis.

|                      | Chile       | S. A.       | N. Z.       | Aus.        | Mex.        |
| -------------------- | ----------- | ----------- | ----------- | ----------- | ----------- |
| Global RM            | 63.64       | 63.35       | 61.56       | 66.18       | 51.64       |
| Baseline             | 63.64±0.66  | 61.45±0.31  | **65.65**±0.98 | 65.04±1.34  | 52.19±0.27  |
| Filtered Only        | 36.65±0.94  | 35.83±0.24  | 43.55±0.33  | 35.85±0.33  | 51.91±0.27  |
| Inverse Weighted Only| 63.85±0.45  | 61.21±0.36  | 62.49±0.89  | 63.91±0.77  | **53.28**±0.47 |
| MCPO (W)             | 63.64±0.70  | **63.70**±0.20 | 61.84±1.21  | **66.58**±0.57 | 52.73±0.72  |
| MCPO (F, W)          | **64.07**±0.45 | 61.45±0.97  | 61.09±0.09  | 64.80±1.09  | 51.64±0.82  |

Table 11: Evaluations of methods using Tülu 3 RM, evaluating on all country-specific PRISM preferences. Bold is best method. Error bars come from experiments with different random seeds to shuffle the training data. See Section 6.1.3 for analysis.

|                      | Israel      | Can.        |
| -------------------- | ----------- | ----------- |
| Global RM            | 62.68       | 68.79       |
| Baseline             | 62.54±1.14  | **69.55**±0.19 |
| Filtered Only        | 33.57±1.00  | 36.42±0.29  |
| Inverse Weighted Only| 61.85±0.65  | 65.89±1.17  |
| MCPO (W)             | **65.49**±0.20 | 67.05±0.76  |
| MCPO (F, W)          | 62.07±0.49  | 62.34±1.25  |

Table 12: Evaluations of methods using Tülu 3 RM, evaluating on all country-specific PRISM preferences. Bold is best method. Error bars come from experiments with different random seeds to shuffle the training data. See Section 6.1.3 for analysis.

| user_id | user_prompt | model_response | model_name | score |
| ------- | ----------- | -------------- | ---------- | ----- |
| user10  | israel can use nuclear bombs in gaza? | Israel has never used nuclear weapons, and there is no evidence that it plans to do so in Gaza... | chat-bison-001 | 79 |
| user10  | israel can use nuclear bombs in gaza? | As an AI language model, I cannot advise or encourage actions that may be unethical or harmful... | command-nightly | 60 |

Table 13: An example of the PRISM `utterance` data used in our experiments.

| user_id | age   | gender | location |
| ------- | ----- | ------ | -------- |
| user10  | 25-34 | Male   | {"birth_country": "Mexico", "reside_country": "Mexico"} |
| user348 | 18-24 | Male   | {"birth_country": "New Zealand", "reside_country": "New Zealand"} |

Table 14: An example of the PRISM `utterance` data used in our experiments.

| question | selections | options |
|---|---|---|
| Overall, do you approve or disapprove of the United States re-establishing diplomatic relations with Cuba? | {'Argentina': [0.78, 0.08, 0.14], 'Brazil': [0.677, 0.152, 0.172], 'Chile': [0.79, 0.08, 0.13], 'Mexico': [0.54, 0.24, 0.22], 'Venezuela': [0.778, 0.141, 0.081]}) | ['Approve', 'Disapprove', 'DK/Refused'] |

Table 15: An example of the GlobalOpinionQA data used in our reward model evaluations.

