# OpenReview forum: "Multi-Cultural Preference Optimization of Reward Models"
_ICLR.cc/2026/Conference — Submitted to ICLR 2026_

### Official Review · Reviewer_Wi4Z · 2025-10-29

**Soundness:** 1
**Presentation:** 2
**Contribution:** 1
**Rating:** 2
**Confidence:** 4

**Summary:**

The paper's goal is to train country-specific reward models for pluralistic alignment. They propose a filtering and weighting step where country-specific preferences that align with a global reward model (RM) are filtered and then the remaining examples are weighted such examples where there is a misalignment with the global RM are given lower weight. They evaluate predictive accuracy on the PRISM dataset and on GlobalOpinionQA.

**Strengths:**

The goal of creating country-specific reward models, as motivated by pluralistic alignment, is an important one. However, unfortunately, I don't believe this paper makes a large enough contribution on either the method or evaluation.

**Weaknesses:**

# Method

The primary method proposed by the authors involves a two-step process: first, they filter out country-specific preferences that are closely aligned with a global reward model (RM). Next, they assign weights to the remaining examples, giving lower weight to those that are misaligned with the global RM.

However, this approach appears somewhat contradictory. Initially, examples that strongly agree with the RM are removed, but in the subsequent step, higher weights are assigned to examples with greater agreement. This seems inconsistent, as step two appears to counteract the effect of step one. Indeed, in most cases, just the weighting on its own performs better. Despite this, the authors claim—both in the abstract and throughout the paper—that their method is "up to 3x more training data-efficient." This claim is based on the idea that filtering increases efficiency by reducing the amount of data, but it is misleading, since the weighting method alone generally performs best.

On its own, this method does not represent a substantial enough contribution to justify acceptance. The approach lacks theoretical justification, so one would expect strong empirical results to compensate; unfortunately, the evaluations presented are also weak.

# Evaluation

## Experimental set-up
The authors evaluate predictive accuracy of their RMs on held-out test sets from the PRISM dataset and on GlobalOpinionQA.

The PRISM dataset has a couple of problems. First, the authors motivate their work in the context of pluralistic alignment and wanting to create models that can steer to different country's values. However, it has been shown that highly salient differences in values between cultures cannot be learned from the PRISM dataset:

> Cultivating Pluralism In Algorithmic Monoculture: The Community Alignment Dataset. Zhang et al 2025 https://arxiv.org/abs/2507.09650

Second, the PRISM dataset is very small. The median number of comparisons per annotator in PRISM is 19. For the countries the authors study, there are between 47-91 annotators. An alternative explanation for why the weighting that they propose is helpful could simply be that it removes outliers from this small and noisy dataset. It's not clear to me that the authors' method would hold up on larger datasets, e.g., the Community Alignment dataset listed above, which has preferences from 5 different countries, each with ~400-1000 annotators and a median of 67 comparisons per annotator.

Regarding GlobalOpinionQA, the authors do not engage with any of the literature calling into question the robustness of survey-based assessments of LLMs, e.g.,

> "Political Compass or Spinning Arrow? Towards More Meaningful Evaluations for Values and Opinions in Large Language Models." Röttger, Hoffman, et al. ACL 2024.

> "Questioning the survey responses of large language models" Dominguez-Olmedo et al. NeurIPS 2024.

> "Randomness, Not Representation: The Unreliability of Evaluating Cultural Alignment in LLMs" Khan et al. FAccT 2025.

## Results

For PRISM, the authors omit error bars from the main text, including them only in the appendix. It is unclear why this choice was made. I recommend reducing the table font size in the main text to accommodate the error bars and ensure they are included. Once the error bars are considered, the author's method does not produce a consistent improvement. For GlobalOpinionQA, there are no error bars, and the authors do not report results for where their method did not have the best performance on PRISM. This may result in a biased understanding of the impact of their method.

**Questions:**

How is the baseline "inverse weighted" method different from MCPO (weighted only) ?

---

### Official Review · Reviewer_3ZWR · 2025-10-30

**Soundness:** 2
**Presentation:** 2
**Contribution:** 2
**Rating:** 4
**Confidence:** 3

**Summary:**

The paper proposes Multi-Cultural Preference Optimization (MCPO), a method for training culturally aware reward models (RMs) that better reflect minority group preferences while maintaining global consistency. MCPO combines two mechanisms: (1) filtering, which removes preference pairs that have a large reward margin between the preferred and dispreferred responses according to the global RM, and (2) weighting, which assigns lower weights to preference pairs that are more inconsistent between the global RM’s judgments and human annotations. Experiments on PRISM and GlobalOpinionQA datasets across seven countries show that MCPO achieves modest accuracy improvements and improved data efficiency (about 3×) compared to standard fine-tuning of RMs.

**Strengths:**

- The paper tackles a timely and socially relevant problem, addressing cultural and demographic diversity in language model alignment within the framework of reward model training.
- The proposed combination of filtering and weighting guided by a global RM is a simple yet original idea for improving cultural specificity while maintaining general alignment.
- The experimental setup uses recent multi-country datasets (PRISM, GlobalOpinionQA), providing a concrete evaluation framework for pluralistic alignment.

**Weaknesses:**

- The paper evaluates only the reward model itself without demonstrating downstream effects on policy alignment or generation quality. While it is reasonable as a first step, given the relatively small gains observed at the RM level, it is unclear whether MCPO-trained RMs would lead to any tangible improvement when used to align an LLM.

- Related to the above, the baselines are limited, primarily GPO, and thus it remains uncertain how significant the advantage of MCPO actually is.

- The paper introduces a filtering threshold τ (Eq. 1) to select minority preference pairs but does not specify its value or provide any ablation on its effect, apart from the implicit extreme case (τ=1; no filtering, represented by MCPO(W) in Tables 3–5). Since τ directly controls the proportion of data retained (and hence the claimed 3× data efficiency), the lack of sensitivity analysis leaves uncertainty about how robust the reported gains are to this hyperparameter choice.

- The filtering step is empirically tested through ablation (Tables 3–5), showing that filtering alone degrades performance while the combination with weighting yields marginal gains. This indicates that filtering may help stabilize training or improve data efficiency, but its theoretical justification remains unclear: the paper provides no formal reasoning why discarding globally consistent pairs should improve cultural alignment beyond empirical evidence.

**Questions:**

The paper frequently refers to “extremeness” of preference samples and provides illustrative examples (Tables 1–2), but a precise mathematical definition is never given. It remains unclear how these values are computed from the reward model scores, or how they relate to the weighting term in Eq. (2). Could the authors clarify how “extremeness” is formulated and how it affects the weighting mechanism?

**Details Of Ethics Concerns:**

No specific ethics concerns.
The work focuses on methodological aspects of reward model training using publicly available datasets (PRISM, GlobalOpinionQA).

---

### Official Review · Reviewer_jVNT · 2025-11-01

**Soundness:** 2
**Presentation:** 2
**Contribution:** 2
**Rating:** 2
**Confidence:** 4

**Summary:**

In this paper, the authors propose Multi-Cultural Preference Optimization (MCPO), a method for training reward models to capture diverse preferences across cultural groups. Experiments on two datasets and two reward models demonstrate that MCPO effectively helps the model learn the preferences of minority communities.

**Strengths:**

1. The problem of multi-cultural optimization is interesting and relevant to the community, as it can guide LLMs toward greater cultural awareness when serving people from diverse backgrounds.
2. The paper introduces a new optimization algorithm that reweights preferences from different cultural groups, improving the reward model’s performance in capturing minority preferences.

**Weaknesses:**

1. The improvement on the whole test set achieved by the proposed method is quite limited compared to the baseline (around 1% in accuracy), which may be negligible given the variance in experimental settings. Moreover, the evaluation does not report the performance on the majority group, which, according to the paper’s definition, should correspond to the U.S. or U.K. It remains unclear whether the model has become overly biased toward minority preferences.
2. The proposed approach appears to be more of a data sampling strategy than a novel training algorithm. The description of the method lacks sufficient clarity to justify why it should work. For instance, why are the more similar preferences from other countries weighted more heavily rather than the more extreme ones?
3. The writing omits several important details, such as the definition of minority preferences and the calculation of “extremeness".

**Questions:**

1. How is extremeness calculated in Table 2?
2. What is the reason for weighting more similar preferences from other countries more heavily, rather than the more extreme ones?
3. Did you use the trained reward model for RLHF fine-tuning of the LLM to evaluate its practical usefulness?

---

### Official Review · Reviewer_tKfB · 2025-11-01

**Soundness:** 2
**Presentation:** 3
**Contribution:** 3
**Rating:** 4
**Confidence:** 3

**Summary:**

The paper proposes **MCPO (Multi-Cultural Preference Optimization)**, a training procedure for **culture-specific** alignment meant to better reflect minority (sub-community) preferences while avoiding skew toward extreme opinions. MCPO combines:
(i) Filtering, (ii) Weighting: a strategy to allocate weights to samples based on their specificity to the considered country.
Extensive experiments on **PRISM** (7 countries: Chile, South Africa, New Zealand, Australia, Mexico, Israel, Canada) and **GlobalOpinionQA (GQA)** demonstrate the effectiveness of the approach compared to baselines while also offering better data-efficiency.

**Strengths:**

**Important topic & clear goal.** Pluralistic alignment is an increasingly important topic given the wide distribution of a small number of models to large numbers of groups and populations. The authors address it by targeting *steerable* country-specific RMs that can plug into RLHF pipelines. The motivation, to represent minority preferences without over-indexing on extremess, is well argued.

**Thorough experiments.** Performed on different models, datasets, comparing against different baselines, and exploring ablations on the methods. The claim are well supported by the experiments.

**Clarity.** The article is easy to follow and offers clear explanations.

**Weaknesses:**

**Novelty/positioning are underspecified.**
The paper cites weighting/filtering precedents (e.g., RAFT, OPTune, Mallows-DPO) but a more precise comparison, especially to GRPO/GPO, would clarify novelty and conditions where MCPO is preferable.

**"Extremeness" lacks a formal definition.**
The core term guiding weights is not mathematically defined; only examples are provided (Tables 1–2) along with labels like "Inappropriate Language," "Malformed Response," "Unwarranted Opinions," and a scalar "extremeness," but the computation rule is absent. This is critical because the **weighting objective (Eq. 4)** depends on it, and the normative framing risks embedding a moral prior. Filtering cultural preferences through a determined moral framework can remove all the specificities of that culture. Please provide an explicit formula and annotation/provenance for *misalignment/extremeness* attributes.

**No harm analysis.**
Since MCPO intentionally moves models toward country-specific preferences, the system risks entrenching harmful or exclusionary norms. The paper gestures at "core global knowledge and values," but offers no empirical *safeguard* or red-teaming results to demonstrate that weighting truly suppresses harmful extremes.

**Filtering may remove useful signal.**
Section 4.1 removes pairs that agree with the global RM to "remove generic, universal preferences." However, *country-salient* but non-controversial norms might also be valuable for stable country-RMs. Empirically, **Filtered-Only** often underperforms (Tables 3 & 5). A clearer justification and ablation (vary $ \tau $ , keep a fraction of "agreeing" pairs) are needed.
Also, any fine-tuning could steer the model away from these "generic, universal training preferences".

**Role of the global RM.**
The **global RM may be the same model as the starting checkpoint** for the minority RM (Fig. 1 note). This raises concerns and requires to analyse whether using the *same* backbone for filtering/weighting biases which pairs survive and how that impacts generalization (e.g., swap in a distinctly trained global RM and compare).

**Scope and reporting gaps on PRISM.**
Evaluation results on the whole PRISM dataset (across all countries) are missing. This would help contextualizing "balanced" alignment.

**Ambiguity in the trade-off target.**
Fig. 3 circles an "optimal" model without defining the frontier criterion that makes that point "optimal." Please provide a clear optimization target or explanation.

**Questions:**

Could you please report **per-country** training/test pair counts *before and after* filtering, and show CIs for the main tables (you include some in Appendix, but consolidating them in the main paper would help).

In page 4, Section 4.1, "This also has the side-effect of simulating a scenario where minority preferences are highly unique (i.e. 90% of the preferences disagree with the global consensus)." Could you please elaborate on this?

---

### Official Review · Reviewer_zmsK · 2025-11-03

**Soundness:** 2
**Presentation:** 3
**Contribution:** 3
**Rating:** 4
**Confidence:** 3

**Summary:**

This paper introduces MCPO, which is a new reward model training algorithm for incorporating diverse cultural preferences in model alignment. MCPO is built off the core idea of using a global reward model to filter out country-labeled preference pairs that already agree with global judgements and then applying a weighting by closeness in the binary ranking loss so that subtle disagreements are emphasized while extreme ones are down-weighted. Experiments are conducted on PRISM and GlobalOpinionQA datasets which show gains over country-specific fine-tuning. The method is claimed to be 3x as data-efficient as fine-tuning due to targeted data filtering.

**Strengths:**

S1. The authors tackle a clear problem within the pluralistic alignment research area around steerability for minority views. This paper positions MCPO alongside other recent groups-focused methods like GPO/GRPO and clearly articulates the questions about balancing opinions, reusing global reward model scores, and identifying the most important parts of any given dataset. The use of PRISM and GlobalOpinionQA are relevant for this purpose.

S2. This work presents a simple and reusable pipeline. There is a great advantage in only requiring a single public global RM for scoring initially and this should make it easy to reproduce with other datasets/benchmarks.

S3. Ablations across the two reward models, OpenAssistant Deberta and Tulu3 are helpful as well as the breakdown of the different parts of MCPO including filtering and weighting both standalone and then combined. This shows the brittleness of the filtering step in the current evaluations.

**Weaknesses:**

W1. Selection-by-disagreement can entrench global RM bias. Filtering pairs that agree with the global RM and up-weighting subtle disagreements presumes the global RM is an unbiased surrogate. If the global RM is itself Western/majority-biased, both filtering and weights can systematically discard or down-scale legitimate minority preferences. Other than ablating across two different existing reward models (which show quite a range of results) this challenge is not addressed

W2. The evaluation construct partly conflates the metric with the mechanism. The “true country-specific” test set is defined by disagreement with the same global RM used for filtering/weighting, creating a dependence between selection and training signal. Although accuracy is still measured against human labels, this coupling complicates interpretation (low accuracy on the subset is claimed as “less biased”). As a result the paper lacks a bias metric or calibration study to counterbalance the claims.

W3. The paper does not present compelling statistical evidence for the benefit of MCPO over GPO and baselines. On the PRISM full test average improvements over the full-data baseline are small (e.g. +0.46 points vs Tulu3 RM), and filtering seemed to hurt more with the tulu3 RM while per-country gains vary widely. The paper does not test significance.

W4. The data-efficiency claims are not well grounded. Providing learning curves over different levels of data use would be a more effective way of validating beyond noting that one-third of data remains after filtering. Since thresholds are configurable by the developer it would make sense to include them in the ablations presented or tackle them separately in the appendix.

**Questions:**

Q1. An open problem in this space is the identification of where ‘minority alignment’ matters and where it doesn’t for users. The authors attempt to solve this by separately evaluating on ‘all’ country-specific preferences vs ‘true’ preferences that have been filtered using the Bradley-Terry model with scores from the global reward model. However this approach doesn’t seem to be validated after the fact. Did the authors consider any human or benchmark evaluations as guardrails for maintaining ‘global preferences’ or other investigations into prompt-topic or other dimensions?

Q2. Could the authors clarify whether any safety filters were used when computing extremeness and when training RMs, to avoid overfitting to harmful text present in low-extremeness pairs? It’s not clear to me that extremeness would be directly correlated with ‘inappropriate language’ or other unsafe responses except from the sampled responses shown by the tables. A small analysis with an open-source safety judge could help understand this.

Q3. Could the authors include a version of Table 6 with the Tulu3 RMs?

Given that this paper tackles such an important problem and does show some promising results, addressing these questions would help me reconsider my score.

---

> ### Comment · Reviewer_zmsK · 2025-11-13
>
> Based on my reading of the other reviews I'm not sure there is an immediate path to acceptance for this work as it stands. However I want to re-iterate that this is an important area of research and it seems the direction the authors have taken is an interesting one! If the authors have follow-ups on the questions or weaknesses identified that would help them in iterating for future submissions I'm happy to continue to answer further questions.

---

### Meta-Review · Area_Chair_tnZM · 2026-01-07

**Summary:**

This paper presents MCPO, a novel reward model training algorithm designed to develop globally-aware LLM alignment models that accurately incorporate and balance diverse cultural preferences from different sub-communities. The authors did not address the significant concerns and weaknesses raised by the reviewers. Therefore, the AC recommends rejection.

**Reviewer Concerns:**

The author did not submit any discussion or rebuttal. So all problems are still unresolved.

**Reviewer Scores:**

The author did not submit any discussion or rebuttal. Thus it is not likely that any reviewer would have changed their scores.

---

### Decision · Program_Chairs · 2026-01-26

Reject